# Linear Feature Encoding for Reinforcement Learning

**Zhao Song, Ronald Parr[†], Xuejun Liao, Lawrence Carin**
Department of Electrical and Computer Engineering
[†] Department of Computer Science
Duke University, Durham, NC 27708, USA

## Abstract

Feature construction is of vital importance in reinforcement learning, as the quality of a value function or policy is largely determined by the corresponding features. The recent successes of deep reinforcement learning (RL) only increase the importance of understanding feature construction. Typical deep RL approaches use a linear output layer, which means that deep RL can be interpreted as a feature construction/encoding network followed by linear value function approximation. This paper develops and evaluates a theory of linear feature encoding. We extend theoretical results on feature quality for linear value function approximation from the uncontrolled case to the controlled case. We then develop a supervised linear feature encoding method that is motivated by insights from linear value function approximation theory, as well as empirical successes from deep RL. The resulting encoder is a surprisingly effective method for linear value function approximation using raw images as inputs.

## 1   Introduction

Feature construction has been and remains an important topic for reinforcement learning. One of the earliest, high profile successes of reinforcement learning, TD-gammon [1], demonstrated a huge performance improvement when expert features were used instead of the raw state, and recent years have seen a great deal of practical and theoretical work on understanding feature selection and generation for linear value function approximation [2–5].

More recent practical advances in deep reinforcement learning have initiated a new wave of interest in the combination of neural networks and reinforcement learning. For example, Mnih et al. [6] described a reinforcement learning (RL) system, referred to as Deep Q-Networks (DQN), which learned to play a large number of Atari video games as well as a good human player. Despite these successes and, arguably because of them, a great deal of work remains to be done in understanding the role of features in RL. It is common in deep RL methods to have a linear output layer. This means that there is potential to apply the insights gained from years of work in linear value function approximation to these networks, potentially giving insight to practitioners and improving the interpretability of the results. For example, the layers preceding the output layer could be interpreted as feature extractors or encoders for linear value function approximation.

As an example of the connection between practical neural network techniques and linear value function approximation theory, we note that Oh et al. [7] introduced spatio-temporal prediction architectures that trained an action-conditional encoder to predict next states, leading to improved performance on Atari games. Oh et al. cited examples of next state prediction as a technique used in neural networks in prior work dating back several decades, though this approach is also suggested by more recent linear value function approximation theory [4].

In an effort to extend previous theory in a direction that would be more useful for linear value function approximation and, hopefully, lead to greater insights into deep RL, we generalize previous work

on analyzing features for uncontrolled linear value function approximation [4] to the controlled case. We then build on this result to provide a set of sufficient conditions which guarantee that encoded features will result in good value function approximation. Although inspired by deep RL, our results (aside from one negative one in Section 3.2 ) apply most directly to the linear case, which has been empirically explored in Liang et al. [8]. This implies the use of a rich, original (raw) feature space, such as sequences of images from a video game without persistent, hidden state. The role of feature encoding in such cases is to find a lower dimensional representation that is suitable for linear value function approximation. Feature encoding is still needed in such cases because the raw state representation is so large that it is impractical to use directly.

Our approach works by defining an encoder and a decoder that use a lower dimensional representation to encode and predict both reward and next state. Our results differ from previous results [4] in linear value function approximation theory that provided sufficient conditions for good approximation. Specifically, our results span two different representations, a large, raw state representation and a reduced one. We propose an efficient coordinate descent algorithm to learn parameters for the encoder and decoder. To demonstrate the effectiveness of this approach, we consider the challenging (for linear techniques) problem of learning features from raw images in pendulum balancing and blackjack. Surprisingly, we are able to discover good features and learn value functions in these domains using just linear encoding and linear value function approximation.

## 2   Framework and Notation

Markov Decision Processes (MDPs) can be represented as a tuple $\langle \mathcal{S}, \mathcal{A}, R, P, \gamma \rangle$, where $\mathcal{S} = \{s_1, s_2, \ldots, s_n\}$ is the state set, $\mathcal{A} = \{a_1, a_2, \ldots, a_m\}$ is the action set, $R \in \mathbb{R}^{nm \times 1}$ represents the reward function whose element $R(s_i, a_j)$ denotes the expected immediate reward when taking action $a_j$ in state $s_i$, $P \in \mathbb{R}^{nm \times n}$ denotes the transition probabilities of underlying states whose element $P\big[(s_i, a), s_j\big]$ is the probability of transiting from state $s_i$ to state $s_j$ when taking an action $a$, and $\gamma \in [0, 1)$ is the discount factor for the future reward. The policy $\pi$ in an MDP can be represented in terms of the probability of taking action $a$ when in state $s$, i.e., $\pi(a|s) \in [0, 1]$ and $\sum_a \pi(a|s) = 1$.

Given a policy $\pi$, we define $P^\pi \in \mathbb{R}^{nm \times nm}$ as the transition probability for the state-action pairs, where $P^\pi(s', a'|s, a) = P\big[(s, a), s'\big] \pi(a'|s')$. For any policy $\pi$, its $Q$-function is defined over the state-action pairs, where $Q^\pi(s, a)$ represents the expected total $\gamma-$discounted rewards when taking action $a$ in state $s$ and following $\pi$ afterwards. For the state-action pair $(s, a)$, the $Q-$function satisfies the following Bellman equation:

$$Q^\pi(s, a) = R(s, a) + \left[\gamma \sum_{s', a'} P^\pi(s', a'|s, a) \, Q^\pi(s', a')\right] \tag{1}$$

### 2.1   The Bellman operator

We define the Bellman operator $T^\pi$ on the $Q-$functions as

$$(T^\pi Q)(s, a) = R(s, a) + \left[\gamma \sum_{s', a'} P^\pi(s', a'|s, a) \, Q(s', a')\right].$$

$Q^\pi$ is known to be a fixed point of $T^\pi$, i.e., $T^\pi Q^\pi = Q^\pi$. Of particular interest in this paper is the *Bellman error* for an approximated $Q$-function to $Q^\pi$, specifically $\mathrm{BE}(\widehat{Q}^\pi) = T^\pi \widehat{Q}^\pi - \widehat{Q}^\pi$. When the Bellman error is 0, the $Q$-function is at the fixed point. Otherwise, we have [9]:

$$\|\widehat{Q}^\pi - Q^\pi\|_\infty \le \|\widehat{Q}^\pi - T^\pi \widehat{Q}^\pi\|_\infty / (1 - \gamma),$$

where $\|\mathbf{x}\|_\infty$ refers to the $\ell_\infty$ norm of a vector $\mathbf{x}$.

### 2.2   Linear Approximation

When the $Q$-function cannot be represented exactly, we can approximate it with a linear function as $\widehat{Q}^\pi(s, a) = \Phi \mathbf{w}^\pi_\Phi$, with $\Phi = [\Phi(s_1, a_1) \ldots \Phi(s_n, a_m)]^T \in \mathbb{R}^{nm \times km}$, $\Phi(s_i, a_j) \in \mathbb{R}^{km \times 1}$ is a feature vector for state $s_i$ and action $a_j$, superscript $T$ represents matrix transpose, and $\mathbf{w}^\pi_\Phi \in \mathbb{R}^{km \times 1}$ is the weight vector.

Given the features $\Phi$, the *linear fixed point* methods [10–12] aim to estimate $\mathbf{w}_\Phi^\pi$, by solving the following fixed-point equation:

$$\Phi\mathbf{w}_\Phi^\pi = \Pi(R + \gamma P^\pi \Phi\mathbf{w}_\Phi^\pi) \qquad (2)$$

where $\Pi = \Phi(\Phi^T\Phi)^{-1}\Phi^T$ is the orthogonal $\ell_2$ projector on $span(\Phi)$. Solving (2) leads to the following linear fixed-point solution:

$$\mathbf{w}_\Phi^\pi = (\Phi^T\Phi - \gamma\Phi^T P^\pi\Phi)^{-1} \Phi^T R.$$

## 2.3 Feature Selection/Construction

There has been great interest in recent years in automating feature selection or construction for reinforcement learning. Research in this area has typically focused on using a linear value function approximation method with a feature selection wrapper.

Parr et al. [2] proposed using the Bellman error to generate new features, but this approach did not scale well in practice. Mahadevan and Maggioni [3] explored a feature generation approach based upon the Laplacian of a connectivity graph of the MDP. This approach has many desirable features, though it did not connect directly to the optimization problem implied by the MDP and could produce worthless features in pathological cases [4].

Geramifard et al. [13] and Farahmand and Precup [14] consider feature construction where features are built up through composition of base or atomic features. Such approaches are reminiscent of classical approaches to features construction. They can be useful, but they can also be myopic if the needed features are not reachable through chains of simpler features where each step along the chain is a demonstrable improvement.

Feature selection solves a somewhat different problem from feature construction. Feature selection assumes that a reasonable set of candidate features are presented to the learner, and the learner's task is to find the good ones from a potentially large set of mostly worthless or redundant ones. LASSO [15] and Orthogonal Matching Pursuit (OMP) [16] are methods of feature selection for regression that have been applied to reinforcement learning [17, 5, 18, 19]. In practice, these approaches do require that good features are present within the larger set, so they do not address the question of how to generate good features in the first place.

## 3 Theory for Feature Encoding

Previous work demonstrated an equivalence between linear value function approximation and linear model approximation [20, 21, 4], as well as the relationship between errors in the linear model and the Bellman error for the linear fixed point [4]. Specifically, low error in the linear model could imply low Bellman error in the linear fixed point approximation. These results were for the uncontrolled case. A natural extension of these results would be to construct features for action-conditional linear models, one for each action, and use those features across multiple policies, i.e., through several iterations of policy iteration. Anecdotally, this approach seemed to work well in some cases, but there were no theoretical results to justify it. The following example demonstrates that features which are sufficient for perfect linear action models and reward models, may not suffice for perfect linear value function approximation.

**Example 1.** *Consider an MDP with a single feature $\phi(x) = x$, two actions that have no effect, $p(x|x, a_1) = 1.0$ and $p(x|x, a_2) = 1.0$, and with $R(x, a_1) = x$ and $R(x, a_2) = -x$. The single feature $\phi$ is sufficient to construct a linear predictor of the expected next state and reward. However, the value function is not linear in $\phi$ since $V^*(x) = |x| / (1 - \gamma)$.*

The significance of this example is that existing theory on the connection between linear models and linear features does not provide sufficient conditions on the quality of the features for model approximation that would ensure good value function approximation for all policies. Existing theory also does not extend to provide a connection between the model error for a set of features and the Bellman error of a Q-function based on these features. To make this connection, the features must be thought of as predicting not only expected next features, but expected next feature-action combinations. Below, we extend the results of Parr et al. [4] to Q-functions and linear state-action models.

**The linear model** Similar to Parr et al. [4], we approximate the reward $R$ and the expected *policy-conditional* next feature $P^\pi \Phi$ in the controlled case, using the following linear model:

$$\hat{R} = \Phi r_\Phi = \Phi(\Phi^T \Phi)^{-1} \Phi^T R \tag{3a}$$

$$\widehat{P^\pi \Phi} = \Phi P_\Phi^\pi = \Phi(\Phi^T \Phi)^{-1} \Phi^T P^\pi \Phi. \tag{3b}$$

Since $\widehat{Q}^\pi = \Phi \mathbf{w}$ for some $\mathbf{w}$, the fixed-point equation in (1) becomes

$$\Phi \mathbf{w} = \Phi r_\Phi + \gamma \Phi P_\Phi^\pi \mathbf{w} \tag{4a}$$

$$\mathbf{w} = (I - \gamma P_\Phi^\pi)^{-1} r_\Phi \tag{4b}$$

**Lemma 2.** *For any MDP M with features $\Phi$ and policy $\pi$ represented as the fixed point of the approximate $Q-$function, the linear-model solution and the linear fixed-point solution are the same.*

**Proof:** See Supplemental Materials. □

To analyze the error in the controlled case, we define the Bellman error for the state-value function, given a policy $\pi$ as

$$\mathrm{BE}\big(\widehat{Q}^\pi(s,a)\big) = R(s,a) + \left[ \gamma \sum_{s',a'} P^\pi(s',a'|s,a)\widehat{Q}^\pi\big(s',a'\big) \right] - \widehat{Q}^\pi(s,a).$$

As a counterpart to Parr et al. [4], we introduce the following reward error and *policy-conditional* per-feature error, in the controlled case as

$$\Delta_R = R - \hat{R} = R - \Phi r_\Phi \tag{5a}$$

$$\Delta_\Phi^\pi = P^\pi \Phi - \widehat{P\Phi}_\pi = P^\pi \Phi - \Phi P_\Phi^\pi. \tag{5b}$$

**Theorem 3.** *For any MDP M with feature $\Phi$, and policy $\pi$ represented as the fixed point of the approximate $Q-$function, the Bellman error can be represented as*

$$BE\big(\widehat{Q}^\pi\big) = \Delta_R + \gamma \Delta_\Phi^\pi \mathbf{w}_\Phi^\pi.$$

**Proof:** See Supplemental Materials. □

Theorem 3 suggests a sufficient condition for a good set of features: If the model prediction error $\Delta_\Phi^\pi$, and reward prediction error $\Delta_R$ are low, then the Bellman error must also be low. Previous work did not give an in-depth understanding of how to construct such features. In Parr et al. [2], the Bellman error is defined only on the training data. Since it is orthogonal to the span of the existing features, there is no convenient way to approximate it, and the extension to off-sample states is not obvious. They used locally weighted regression with limited success, but the process was slow and prone to the usual perils of non-parametric approximators, such as high sensitivity to the distance function used.

One might hope to minimize (5a) and (5b) directly, perhaps using sampled states and next states, but this is not a straightforward optimization problem to solve in general, because the search space for $\Phi$ is the space of functions and because $\Phi$ appears inconveniently on both sides of 5(b) making it difficult rearrange terms to solve for $\Phi$ as an optimization problem with a fixed target. Thus, without additional assumptions about how the states are initially encoded and what space of features will be searched, it is challenging to apply Theorem 3 directly. Our solution to this difficulty is to apply the theorem in a somewhat indirect manner: First we assume that the input is a rich, *raw* feature set (e.g., images) and that the primary challenge is reducing the size of the feature set rather than constructing more elaborate features. Next, we restrict our search space for $\Phi$ to the space of linear encodings of these raw features. Finally, we require that these encoded features are predictive of next *raw* features rather than next encoded features. This approach differs from what Theorem 3 requires but it results in an easier optimization problem and, as shown below, we are able to use Theorem 3 to show that this alternative condition is sufficient to represent the true value function.

We now present a theory of *predictively optimal feature encoding*. We refer to the features that ultimately are used by a linear value function approximation step using the familiar $\Phi$ notation, and we refer to the inputs before feature encoding as the *raw* features, $A$. For $n$ samples and $l$ raw features, we can think of $A$ as an $nm \times lm$ matrix. For every row in $A$, only the block corresponding to the action taken is non-zero. The raw features are operated on by an encoder:

**Definition 4.** *The* encoder, $\mathcal{E}_\pi$ *(or* $E_\pi$ *in the linear case) is a transformation* $\mathcal{E}_\pi(A) = \Phi$. *We use the notation* $\mathcal{E}_\pi$ *because we think of it as* encoding *the raw state. When the encoder is linear,* $\mathcal{E}_\pi = E_\pi$, *where* $E_\pi$ *is an* $lm \times km$ *matrix that right multiplies* $A$, $AE_\pi = \Phi$.

We want to encode a reduced size representation of the raw features sufficient to predict the next expected reward and raw features because, as proven below, doing so is a sufficient (though not necessary) condition for good linear value function approximation. Prediction of next raw feature and rewards is done via a decoder, which is a matrix in this paper, but could be non-linear in general:

**Definition 5.** *The* decoder, $D$, *is a* $km \times (lm + 1)$ *matrix predicting* $[P^\pi A, R]$ *from* $\mathcal{E}_\pi(A)$.

This approach is distinct from the work of Parr et al. [4] for several reasons. We study a set of conditions on a reduced size feature set and study the relationship between the reduced feature set and the original features, and we provide an algorithm in the next section for constructing these features.

**Definition 6.** $\Phi = \mathcal{E}_\pi(A)$ *is* predictively optimal *with respect to $A$ and $\pi$ if there exists a $D_\pi$ such that* $\mathcal{E}_\pi(A)D_\pi = [P^\pi A, R]$.

### 3.1 Linear Encoder and Linear Decoder

In the linear case, a predictively optimal set of features satisfies:

$$AE_\pi D_\pi = AE_\pi[D_\pi^s, D_\pi^r] = [P^\pi A, R] \tag{6}$$

where $D_\pi^s$ and $D_\pi^r$ represent the first $lm$ columns and the last column of $D_\pi$, respectively.

**Theorem 7.** *For any MDP M with predictively optimal* $\Phi = AE_\pi$ *for policy $\pi$, if the linear fixed point for $\Phi$ is* $\widehat{Q}^\pi$, $BE(\widehat{Q}^\pi) = 0$.

**Proof:** See Supplemental Materials. □

### 3.2 Non-linear Encoder and Linear Decoder

One might expect that the results above generalize easily to the case where a more powerful encoder is used. This could correspond, for example, to a deep network with a linear output layer used for value function approximation. Surprisingly, the generalization is not straightforward:

**Theorem 8.** *The existence of a non-linear encoder $\mathcal{E}$ and linear decoder $D$ such that* $\mathcal{E}(A)D = [P^\pi A, R]$ *is* **not** *sufficient to ensure predictive optimality of $\Phi = \mathcal{E}(A)$.*

**Proof:** See Supplemental Materials. □

This negative result doesn't shut the door on combining non-linear encoders with linear decoders. Rather, it indicates that additional conditions beyond those needed in the linear case are required to ensure optimal encoding. For example, requiring that the encoded features lie in an invariant subspace of $P^\pi$ [4] would be a sufficient condition (though of questionable practicality).

## 4 Iterative Learning of Policy and Encoder

In practice we do not have access to $P^\pi$, but do have access to the raw feature representation of sampled states and sampled next states. To train the encoder $E_\pi$ and decoder $D_\pi$, we sample states and next states from a data collection policy. When exploration is not the key challenge, this can be done with a single data collection run using a policy that randomly selects actions (as is often done with LSPI [22]). For larger problems, it may be desirable to collect additional samples as the policy changes. These sampled states and next states are represented by matrices $\tilde{A}$ and $A'$, respectively.

Theorem 7 suggests that given a policy $\pi$, zero Bellman error can be achieved if features are encoded appropriately. Subsequently, the obtained features and resulting Q-functions can be used to update the policy, with an algorithm such as LSPI. In a manner similar to the policy update in LSPI, the non-zero blocks in $A'$ are changed accordingly after a new policy is learned. With the updated $A'$, we re-learn the encoder and then repeat the process, as summarized in Algorithm 1. It may be desirable to update the policy several times while estimating $\widehat{Q}_\pi$ since the encoded features may still be useful if the policy has not changed much. Termination conditions for this algorithm are typical approximate policy iteration termination conditions.

### 4.1 Learning Algorithm for Encoder

In our implementation, the encoder $E_\pi$ and decoder $D_\pi$ are jointly learned using Algorithm 2, which seeks to minimize $\|\tilde{A}E_\pi D_\pi - [A', R]\|_F$ by coordinate descent [23], where $\|X\|_F$ represents the Frobenius norm of a matrix $X$. Note that $\tilde{A}$ can be constructed as a block diagonal matrix, where each block corresponds to the samples from each action. Subsequently, the pseudoinverse of $\tilde{A}$ in Algorithm 2 can be efficiently computed, by operating on the pseudoinverse of each block in $\tilde{A}$.

---

**Algorithm 1** Iterative Learning of Encoder and Policy

**while** Termination Conditions Not Satisfied **do**
    Learn the encoder $E_\pi$ and decoder $D_\pi$
    Estimate $\widehat{Q}^\pi$
    Update the next raw state $A'$, by changing the position of non-zero blocks according to the greedy policy for $\widehat{Q}^\pi$.
**end while**

---

Algorithm 2 alternatively updates $E_\pi$ and $D_\pi$ until one of the following conditions is met: (1) the number of iterations reaches the maximally allowed one; (2) the residual $\frac{\|\tilde{A}E_\pi D_\pi - [A', R]\|_F}{\|[A', R]\|_F}$ is below a threshold; (3) the current residual is greater than the previous residual. For regularization, we use the truncated singular value decomposition (SVD) [24] when taking the pseudo-inverses of $\tilde{A}$ to discard all but the top $k$ singular vectors in each block of $\tilde{A}$.

---

**Algorithm 2** Linear Feature Discovery

LINEARENCODERFEATURES ( $\tilde{A}$, $A'$, $R$, $k$ )
$D_\pi \leftarrow \texttt{rand}(km, lm + 1)$
**while** Convergence Conditions Not Satisfied **do**
    $E_\pi \leftarrow \tilde{A}^\dagger [A', R] D_\pi^\dagger$
    $D_\pi \leftarrow (\tilde{A}E_\pi)^\dagger [A', R]$
**end while**
**return** $E_\pi$

See text for termination conditions.
$\texttt{rand}$ represents samples from uniform $[0, 1]$.
$\dagger$ is the (truncated) Moore-Penrose pseudoinverse.

---

Algorithm 2 is based on a linear encoder and a linear decoder. Consequently, one may notice that the value function is also linear in the domain of the raw features, i.e., the value function can be represented as $\widehat{Q}^\pi = \tilde{A} E_\pi \mathbf{w} = \tilde{A}\mathbf{w}'$ with $\mathbf{w}' = E_\pi \mathbf{w}$. One may wonder, why it is not better to solve for $\mathbf{w}'$ directly with regularization on $\mathbf{w}'$? Although it is impractical to do this using batch linear value function approximation methods, due to the size of the feature matrix, one might argue that on-line approaches such as deep RL techniques approximate this approach by stochastic gradient descent. To the extent this characterization is accurate, it only increases the importance of having a clear understanding of feature encoding as an important sub-problem, since this is the natural interpretation of everything up to the final layer in such networks and is even an explicit objective in some cases [7].

## 5 Experiments

The goal of our experiments is to show that the model of and algorithms for feature encoding presented above are practical and effective. The use of our encoder allows us to learn good policies using linear value function approximation on raw images, something that is not generally perceived to be easy to do. These experiments should be viewed as validating this approach to feature encoding, but not competing with deep RL methods, which are non-linear and use far greater computational resources.

We implement our proposed linear encoder-decoder model and, for comparison, the random projection model in Ghavamzadeh et al. [25]. We tested them on the Inverted Pendulum and Blackjack [26], two popular benchmark domains in RL. Our test framework creates raw features using images, where the elements in the non-zero block of $\tilde{A}$ correspond to an image that has been converted to vector by concatenating the rows of the image. For each problem, we run Algorithm 1 50 times independently to account for the randomness in the training data. Our training data are formed by running a simulation for the desired number of steps and choosing actions at random. For the encoder, the number of features $k$ is selected over the validation set to achieve the best performance. All code is written in MATLAB and tested on a machine with 3.1GHz CPU and 8GB RAM. Our test results show that Algorithm 1 cost at most half an hour to run, for the inverted pendulum and blackjack problems.

To verify that the encoder is doing something interesting, rather than simply picking features from $\tilde{A}$, we also tried a greedy, sparse reinforcement learning algorithm, OMP-TD [5] using $\tilde{A}$ as the candidate feature set. Our results, however, showed that OMP-TD's performance was much worse than the approach using linear encoder. We skip further details on OMP-TD's performance for conciseness.

## 5.1 Inverted Pendulum

We used a version of the inverted pendulum adapted from Lagoudakis and Parr [22], a continuous control problem with 3 discrete actions, left, right, or nothing, corresponding to the force applied to a cart on an infinite rail upon which an inverted pendulum is mounted. The true state is described by two continuous variables, the angle and angular velocity of the pendulum. For the version of the problem used here, there is a reward of $0$ for each time step the pendulum is balanced, and a penalty of $-1$ for allowing the pendulum to fall, after which the system enters an absorbing state with value $0$. The discount factor is set to be $0.95$.

For the training data, we collected a desired number of trajectories with starting angle and angular velocity sampled uniformly on $[-0.2, 0.2]$. These trajectories were truncated after $100$ steps if the pendulum had not already fallen. Algorithm 2 did not see the angle or angular velocity. Instead, the algorithm was given two successive, rendered, grayscale images of the pendulum. Each image has $35 \times 52$ pixels and hence the raw state is a $35 \times 52 \times 2 = 3640$ dimensional vector. To ensure that these two images are a Markovian representation of the state, it was necessary to modify the simulator. The original simulator integrated the effects of gravity and the force applied over the time step of the simulator. This made the simulation more accurate, but has the consequence that the change in angle between two successive time steps could differ from the angular velocity. We forced the angular velocity to match the change in angle per time step, thereby making the two successive images a Markovian state.

We compare the linear encoder with the features using radial basis functions (RBFs) in Lagoudakis and Parr [22], and the random projection in Ghavamzadeh et al. [25]. The learned policy was then evaluated $100$ times to obtain the average number of balancing steps. For each episode, a maximum of $3000$ steps is allowed to run. If a run achieves this maximum number, we claim it as a success and count it when computing the probability of success. We used $k = 50$ features for both linear encoder and random projection.

Figure 1 shows the results with means and 95% confidence intervals, given different numbers of training episodes, where Encoder$-\tau$ corresponds to the version of Algorithm 1 with $\tau$ changes in the encoder. We observe that for most of the points, our proposed encoder achieves better performance

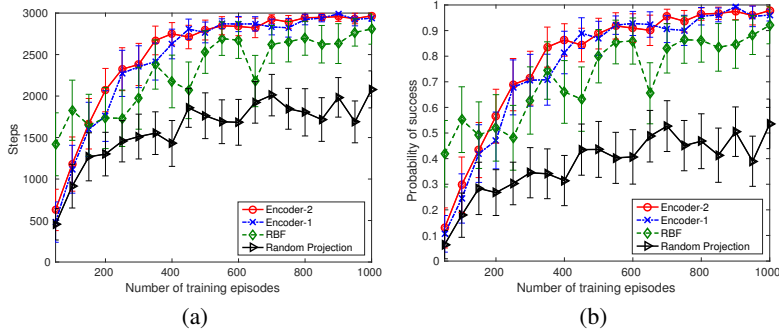

(a)                                              (b)

Figure 1: (a) Number of balancing steps and (b) prob. of success, vs. number of training episodes.

than RBFs and random projections, in terms of both balancing steps and the probability of success. This is a remarkable result because the RBFs had access to the underlying state, while the encoder was forced to discover an underlying state representation based upon the images. Moreover, Encoder$-2$ achieves slightly better performance than Encoder$-1$ in most of the testing points. We also notice that further increasing $\tau$ did not bring any obvious improvement, based on our test.

## 5.2 Blackjack

There are 203 states in this problem, so we can solve directly for the optimal value $V^*$ and the optimal policy $\pi^*$ explicitly. The states from 1-200 in this problem can be completely described by the information from the ace status (usable or not), player's current sum (12-21), and dealer's one showing card (A-10). The terminal states 201- 203 correspond to win, lose, and draw, respectively. We set $k = 203$ features for the linear encoder.

To represent raw states for the encoder, we use three concatenated sampled MNIST digitsand hence a raw state is a $28 \times 28 \times 3 = 2352$ dimensional vector. Two examples of such raw states are shown in Figure 2. Note that three terminal states are represented by "300", "400", and "500", respectively.

The training data are formed by executing the random policy with the desired number of episodes. Our evaluation metrics for a policy represented by the value $V$ and the corresponding action $a$ are

$$\text{Relative value error} = \|V - V^*\|_2 \,/\, \|V^*\|_2, \quad \text{Action error} = \|a - a^*\|_0.$$

We compare the features discovered by the linear encoder and random projection against indicator functions on the true state, since such indicator features should be the gold standard.

We can make the encoder and random projection's tasks more challenging by adding noise to the raw state. Although it is not guaranteed in general (Example 1), it suffices to learn a single encoder that persisted across policies for this problem, so we report results for a single set of encoded features. We denote the algorithms using linear encoder as `Encoder-Image-`$\kappa$ and the algorithms using random projection as `Random-Image-`$\kappa$, where $\kappa$ is the number of possible images used for each digit. For example $\kappa = 10$ means that the image for each digit is randomly selected from the first 10 images in the MNIST training dataset.

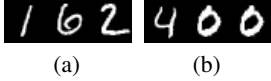

(a)        (b)

Figure 2: Two examples of the blackjack state rendered as three MNIST digits.

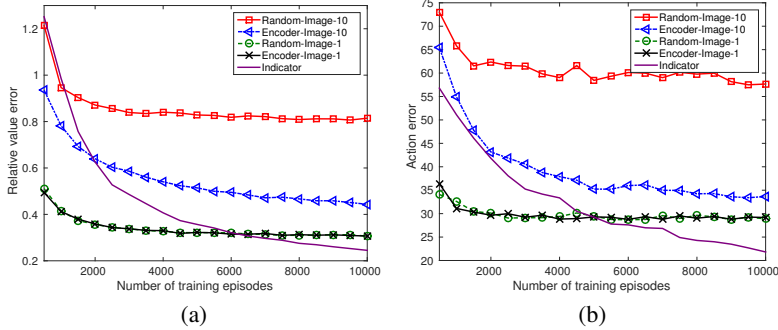

(a)                (b)

Figure 3: (a) Relative value error and (b) action error, as functions of the number of training episodes. An additional plot for the actual return is provided in Supplemental Materials.

Figure 3 shows the surprising result that `Encoder-Image-1` and `Random-Image-1` achieve superior performance to indicator functions on the true state when the number of training episodes is less than or equal to 6000. In this case, the encoded state representation wound up having less than 203 effective parameters because the SVD in the pseudoinverse found lower dimensional structure that explained most of the variation and discarded the rest as noise because the singular values were below threshold. This put the encoder in the favorable side of the bias-variance trade off when training data were scarce. When the number of training episodes becomes larger, the indicator function outperforms the linear encoder, which is consistent with its asymptotically optimal property. Furthermore, the performance of the encoder becomes worse as $\kappa$ is larger. This matches our expectation that a larger $\kappa$ means that a state would be mapped to more possible digits and thus extracting features for the same state becomes more difficult. Finally, we notice that our proposed encoder is more robust to noise, when compared with random projection: `Encoder-Image-10` outperforms `Random-Image-10` with remarkable margins, measured in both relative value error and action error.

## 6 Conclusions and Future Work

We provide a theory of feature encoding for reinforcement learning that provides guidance on how to reduce a rich, raw state to a lower-dimensional representation suitable for linear value function approximation. Our results are most compelling in the linear case, where we provide a framework and algorithm that enables linear value function approximation using a linear encoding of raw images. Although our framework aligns with practice for deep learning [7], our results indicate that future work is needed to elucidate the additional conditions that are needed to extend theory to guarantee good performance in the non-linear case.

**Acknowledgements**

We thank the anonymous reviewers for their helpful comments and suggestions. This research was supported in part by ARO, DARPA, DOE, NGA, ONR and NSF.

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
