[Supplementary Material]

# Linear Feature Encoding for Reinforcement Learning Supplemental Materials

**Zhao Song, Ronald Parr[†], Xuejun Liao, Lawrence Carin**
Department of Electrical and Computer Engineering
[†] Department of Computer Science
Duke University, Durham, NC 27708, USA

## 1   Proof of Lemma 2

**Proof:** We start from the linear model solution and proceed as follows:

$$
\begin{aligned}
\mathbf{w} &= \left(I - \gamma P_\Phi^\pi\right)^{-1} r_\Phi \\
&= \left(I - \gamma (\Phi^T \Phi)^{-1} \Phi^T P^\pi \Phi\right)^{-1} (\Phi^T \Phi)^{-1} \Phi^T R \\
&= \left(\Phi^T \Phi - \gamma \Phi^T P^\pi \Phi\right)^{-1} \Phi^T R = \mathbf{w}_\Phi^\pi.
\end{aligned}
$$

where the penultimate substitutes the definition of $r_\Phi$ and $P_\Phi^\pi$ in (3a) and (3b) of the main text, respectively. $\square$

## 2   Proof of Theorem 3

**Proof:** The Bellman error in the context of linear value functions can be represented as

$$
\mathrm{BE}\big(\widehat{Q}^\pi(s,a)\big) = R(s,a) + \left[\gamma \sum_{s',a'} P^\pi\big(s',a'|s,a\big)\Phi\big(s',a'\big)\mathbf{w}_\Phi^\pi\right] - \Phi(s,a)\mathbf{w}_\Phi^\pi \qquad \text{(A1)}
$$

We proceed to represent (A1) in its corresponding matrix form as

$$
\mathrm{BE}\big(\widehat{Q}^\pi\big) = R + \gamma P^\pi \Phi \mathbf{w}_\Phi^\pi - \Phi \mathbf{w}_\Phi^\pi \qquad \text{(A2)}
$$

Plugging (5) of the main text into (A2), we have

$$
\begin{aligned}
\mathrm{BE}\big(\widehat{Q}^\pi\big) &= R + \gamma P^\pi \Phi \mathbf{w}_\Phi^\pi - \Phi \mathbf{w}_\Phi^\pi \\
&= (\Delta_R + \Phi r_\Phi) + \gamma(\Delta_\Phi^\pi + \Phi P_\Phi^\pi)\mathbf{w}_\Phi^\pi - \Phi \mathbf{w}_\Phi^\pi \\
&= \Delta_R + \gamma \Delta_\Phi^\pi \mathbf{w}_\Phi^\pi + \Phi r_\Phi - \Phi(I - \gamma P_\Phi^\pi)\mathbf{w}_\Phi^\pi \\
&= \Delta_R + \gamma \Delta_\Phi^\pi \mathbf{w}_\Phi^\pi + \Phi r_\Phi - \Phi(I - \gamma P_\Phi^\pi)\mathbf{w} \\
&= \Delta_R + \gamma \Delta_\Phi^\pi \mathbf{w}_\Phi^\pi.
\end{aligned}
$$

The penultimate step follows from Lemma 2, and the last follows equation (4b) of the main text. $\square$

## 3   Proof of Theorem 7

**Proof:** Equation (6) of the main text implies that there exist perfect linear predictors of the reward and the expected next state, given $\Phi = AE_\pi$. Specifically, we pick $P_\Phi^\pi = D_\pi^s E_\pi$ and $r_\Phi = D_\pi^r$. Next, we have

$$
\begin{aligned}
\Delta_\Phi^\pi &= P^\pi \Phi - \Phi P_\Phi^\pi = P^\pi \Phi - AE_\pi D_\pi^s E_\pi \\
&= P^\pi \Phi - P^\pi AE_\pi = P^\pi \Phi - P^\pi \Phi = 0
\end{aligned}
$$

and
$$\Delta_R = R - \Phi r_\Phi = R - A E_\pi D_\pi^r = R - R = 0$$
From Theorem 3, this implies zero Bellman error. □

## 4 Proof of Theorem 8

**Proof:** Consider an MDP for which the $Q$ and $P^\pi$ are *not* linear in $A$. This would be the typical case in which one would wish to use a neural network or other non-linear approximation method. $P^\pi$ can be deterministic so that $P^\pi A$ is a matrix of raw encodings of actual states, not mixtures. Assume $k = l$ and pick $\mathcal{E} = P^\pi$, i.e., pick a vacuous encoder. (For this example we will ignore the reward because predicting the reward does not change anything.) This implies a vacuous decoder $D = I$. When combined, these predict $P^\pi A$. However, $Q$ is not linear in $A$ by assumption and therefore is not linear in $\Phi = \mathcal{E}(A)$ since elements of $\mathcal{E}(A)$ are also elements of $A$. Therefore, a linear value function using features $\mathcal{E}(A)$ may have nonzero Bellman error. □

## 5 Additional Results

After learning a policy $\pi$, we can evaluate $V_\pi$ exactly since there are just 203 states. Subsequently, we have
$$\text{Actual return} = \sum_s V_\pi(s) \, b_0(s),$$

where $b_0$ corresponds to a uniform distribution. Figure A1 shows the actual returns for different algorithms, where the "optimal" curve is obtained by solving the MDP.

Figure A1: Actual return as a function of the number of training episodes, in the Blackjack problem.