[Reviews · NeurIPS 2016]

Reviewer 1

Summary

This paper presents a method that builds a feature representation for Reinforcement Learning with Linear Value Function. More particularly, the method is a supervised linear encoding technique which consists in finding a feature representation such that the Bellman error is low. This is an interesting technique to adapt to LSPI where the Bellman error can be arbitrarily bad with a fixed linear representation which is not the case with this technique. Then, the authors present a way to compute the features thanks to an algorithm that minimizes a well-defined loss. Finally, they present the benefits of their algorithm compared to classical ways of building linear features on RL benchmarks.

Qualitative Assessment

The paper is very well written and easy to follow for the most part. The main idea is to automatically construct a set of features for which the Bellman error is small which is a nice idea as the main problem to analyze an algorithm such as LSPI is the fact that the image of the Q-function for a given policy is not in the linear span of the features which can make the algorithm arbitrarily bad. The authors proposed an interesting algorithm which involves linear encoders and auto-encoders in order to build the sampled matrix of the dynamics and the reward function. However, it would be easier if the authors put an emphasis on the main difficulties of their algorithm which are the calculation of the truncated pseudo-inverse of A, the choice of the number of features k and the choice of the rank in the SVD. Also, it would be nice to add some insights on the convergence (if there is effectively convergence) of the coordinate descent which looks like a fixed-point equation update. Interesting results are presented in the experiments. However, the choice of k the number of features is not well motivated. Besides, the rank of the truncated SVD is not indicated for the inverted pendulum experiment or it is not very clear. Finally, I think that the authors could also compare their algorithm to KLSPI where the features are constructed thanks to a Kernel and an Engel dictionary. Despite those remarks, I do think that the paper presents some interesting perspectives for future works, differentiate itself from features selection and do connections with deep RL where the encoder is non linear.

Confidence in this Review

2-Confident (read it all; understood it all reasonably well)


Reviewer 2

Summary

The authors present a new formalism and algorithm for linear dimensionality reduction/feature encoding for reinforcement learning. They explain how linear feature encoding can be directly connected to bellman error, present an algorithm for learning a good encoding, and present compelling results on two domains using raw images as input.

Qualitative Assessment

This paper appears to be the new state of the art for LSPI type algorithms. The authors begin by explaining what an appropriate feature encoding/construction criterion looks like in this setting. They then give a straightforward algorithm for learning feature encoder and decoders, and they present very impressive empirical results on domains with image inputs that I think many researchers would assume need nonlinear feature construction. The algorithm is straightforward and I expect will become the new standard in these settings. The presentation is thorough and clear, and the paper is enjoyable to read. Minor: "et al. cited" - "et al.\ cited" Algorithm 2 contains a "LinearAutoencoderFeatures" function. Do you really want the "Auto" prefix in light of the comment on line 155?

Confidence in this Review

2-Confident (read it all; understood it all reasonably well)


Reviewer 3

Summary

This paper proposes a novel approach to construct (encode) linear features for reinforcement learning (RL) problems (Algorithm 1 and 2), and demonstrates experiment results in two benchmark examples: inverted pendulum and blackjack (Section 5).

Qualitative Assessment

The paper is very interesting in general. To the best of my knowledge, the proposed feature construction (encoding) algorithm is novel. The experiment results in Section 5 are rigorous and convincing. However, in general I think the paper is borderline for acceptance, for the following reasons: 1) The paper is not well-written in general, and the following parts of the paper are hard to digest: [a] The statements of Lemma 2 and Theorem 3. In particular, it takes me quite a while to locate the "linear-model solution" and the "linear fixed-point solution". [b] The last paragraph of Section 4.1. [c] The last paragraph of Section 5.1, in particular, the definitions of Encoder 1 and 2 are still not clear. 2) There is no theoretical justification of the proposed feature construction algorithm. All the analyses in Section 3 are just motivation for the proposed algorithm. There is no analysis or discussion on *under which scenarios the proposed algorithm is expected to work well or not so well*. I fully understand that a formal analysis on the algorithm performance might be very challenging; but I do believe that the authors should at least discuss (in English) about the performances of the proposed algorithm under various scenarios.

Confidence in this Review

2-Confident (read it all; understood it all reasonably well)


Reviewer 4

Summary

The authors are interested in the problem of what makes a good approximation architecture and feature set, and how to assemble a feature set that has these qualities. First, they show that in fixed-point solutions to Q function approximation, the Bellman error is a sum of model approximation errors - this is a slight extension to Parr et al., which showed it for value functions. They they introduce the idea of an encoder and a decoder, functions used to transform initial state information to a feature set, and then transform these features into model approximations. They show that in the linear case, an encoder which perfectly fits the model gives a Bellman error of 0. The next theorem demonstrates that for a nonlinear encoder and linear decoder, perfect model prediction is not sufficient for feature set optimality (this is perhaps the most interesting result of the paper). Given this guidance, they describe an algorithm to learn an encoder and policy. Lastly, they run this algorithm and demonstrate its ability to learn on benchmark domains, using some unusual state representations (grayscale images of an inverted pendulum, rather than its state directly, or mnist digits representing the blackjack hands).

Qualitative Assessment

For me, this paper is a low accept. Through 3.1, the paper is a small, fairly obvious step beyond Parr et al. (acknowledged by the authors). The thought about non-linear encoders is more interesting, and may have some ramifications on the hot approach of using deep nets for neural networks. The iterative learner is fine, and is interesting enough to be published. Five years ago, I'd complain about the amount of data needed for such training, but in context with the deepmind results, this conversation seems less relevant. The experiments are an interesting twist on the normal benchmarks, and allow for some more interesting encoders. Overall, I wish this paper had said less about the Parr-extended results (I got to page five without learning anything new), and more about the remainder, which was quite interesting. It's a slight toehold on some interesting results about nonlinear representations that are sorely needed.

Confidence in this Review

3-Expert (read the paper in detail, know the area, quite certain of my opinion)


Reviewer 5

Summary

This work aims to develop a linear feature encoder and decoder that are capable of predicting the next reward and next features. The paper includes proofs that show that this representation is sufficient for developing a good linear function approximator. Additionally, empirical results are included that display that this method can work with raw image inputs.

Qualitative Assessment

================== Technical quality ================== The paper does a good job with showing theoretically that linear feature encoding can be good for linear function approximation. However, more justifications and analysis for the empirical results would have been useful. For example, why were RBFs given the true state rather than the raw pixels in the Inverted Pendulum experiment? Why wasn't it used for comparison at all in the Blackjack experiment? Additionally, the results for the encoder and random projections are nearly identical for the Blackjack non-noisy task. It seems then that random projections can also be capable of handling inputs from raw images. Was this expected? Why might it perform well here but not in the inverted pendulum? The main argument that is missing here is why/when the encoding method would perform better. ================== Novelty ================== This work introduces a novel approach for learning good features. Additionally, the paper justifies the use of the feature encoder and decoder and shows that this representation can yield a good function approximator. However, one point that should be included is why this approach is better than the pre-existing approaches. ================== Potential impact ================== This approach gives insight into how to develop good features for linear function approximation, which remains an important problem in reinforcement learning. I believe many researchers would be interested in using a linear feature encoder since its feasibility was justified theoretically and empirically. ================== Clarity ================== This paper was well written and easy to follow. It was very clear from the paper how one might implement this approach. One potentially confusing sentence though was from line 288. It was unclear to me what it would mean to have tau changes in the encoder. ================== Other points ================== Line 91: features -> feature

Confidence in this Review

2-Confident (read it all; understood it all reasonably well)


Reviewer 6

Summary

This paper addresses the problem of designing and learning models for planning in reinforcement learning. More specifically, it proposes a linear model comprising of an "encoder", mapping the observation space into a latent space, and a "decoder", mapping the latent space to the predicted next state and reward. The encoder/decoder model draws inspiration from autoencoders (linear) but couples transition dynamics and reward under the same model. The authors use Parr's (2007) Bellman error decomposition to analyze the their model. A factorization approach is chosen for learning such models using the SVD. Experiments finally show that this model can be used in high dimensional settings when learning directly from pixels.

Qualitative Assessment

Summary: The idea of coupling reward and dynamics in an autoencoder-like model is a novel contribution which could benefit our community. I also appreciate that the authors have applied their model on pixel-based observation spaces. However, I find the theory of lines 124 to 136 unnecessary and the fact that it reproduces Parr (2007) line by line is problematic (more on this below). Also, example 1 seems misguided since it simply does adopt the right problem formulation to start with (it seems sufficient to simply start with a Markov chain over state-action pairs). Detailed comments: Abstract: l. 4. and sect. 1 l.25: "Typical deep RL [...]" and "It is common" Is that true ? Beside DQN, what are other examples ? Is it true for TD-Gammon ? section 1, l. 21: "researchers at Deepmind". I would use a different form. Simply citing the DQN paper would probably be enough. sect 1. l. 31: The qualifier "novel" seems unnecessary. sect 1. l. 41: "Aside from the negative one". We don't know yet at this point what this mean. Perhaps you can reference the section in which the negative one is presented. section 1. l.42 : "Persistent hidden state". This is not clear what you mean here. Perhaps you meant to compare your model-based approach with more of an RNN or PSR-type model. If this is the case, you would have to argue that the proposed model is somehow able to learn a representation which makes the observations as "Markov" as possible, that you can "make state" without state/belief state updating. section 1. l. 47: "Our results differ from". Please cite the work implied in this sentence. Perhaps Parr (2007), Sutton (2008) ? section 2. l. 56: This is not any MDP, it's a *Discounted* MDP. Why do you need to define the reward function over $(n \times m) \times 1$ ? section 2. l. 62: "The policy $\pi$": This is a particular kind of policy, namely a "stochastic" or "randomized" policy. section 2. l. 65: Sum: "Expected total *sum* of $\gamma$-discounted" section 2.1 l. 71: "Otherwise, we have [...]. This sentence lacks context. It should probably be mentioned that the Bellman operator is a contraction and that you solve for the fixed point through value iteration. section 2.3 : "has typically": please cite relevant literature. "Feature selection wrapper": "wrapper" is more of a term that you would use in software engineering. The meaning is not clear in this context. section 2.3 l. 86: Please explain why it did not scale well. section 2.3 l. 88: "connect directly to the optimization problem". This isn't clear. Knowing this paper, I think that you would want to say is that PVFs only consider the transition dynamics and not the reward structure. section 2.3 l. 91: "classical approaches". What are they ? Please cite relevant work. section 2.3 l. 98: it could be relevant to also mention "Value Pursuit Iteration" by Farahmand and Precup. section 3. l. 109: "Anecdotally" generally connotes with "amusing", which is not the intended meaning here. section 3. l. 106-108: I find this sentence confusing because it mixes concepts of "action-conditional" linear models (which is more about the "evaluation problem") with the "control problem" (policy/value iteration). One does not imply the other. section 3. l. 109. "The following example [...]". I find this sentence confusing and it sounds in a way that contradics Parr (2007). Also, you need to define more formally what you mean by "linear action models" earlier in the section. Lines 124 to 136 (and section 3.0 as a whole) seem unnecessary. The derivations follow exactly line by line (and using the same notation) that of Parr (2007). It would suffice to say that in order to represent Q-values, we need to consider the augmented Markov chain over state-action pairs. You define the state $\tilde{s}_t = (s_t, a_t)$ and then simply use Parr's results over this chain. The same argument applies when you go from TD(0) to SARSA(0) : see Sutton & Barto for example https://webdocs.cs.ualberta.ca/~sutton/book/ebook/node64.html section 3. following definition 5: A novel aspect of the proposed approach is that it couples reward and transition dynamics under the same predictor. I think that it's an important design choice for which it would be worth spending more time explaining. You could for example contrasts this approach with one based exclusively on the transitions dynamics (Mahadevan's PVFs for example). section 3.1, l. 170: Theorem numbering jumps from 3 to 7. section 4, l. 208: "termination conditions are typical API [...]", please briefly explain why they still hold. section 4.1 : Please explain why you chose coordinate descent. Also, it would be useful to study the effect of the estimation error in $\tilde{A}$ on the quality of the learned features. Is it stable ? Lines 220-223 is useful to know but it is also more difficult to clearly see what is the main algorithm proposed in this section. It could perhaps be moved to the appendix. section 5, l. 246: "Ambitious". Many examples of large scale linear RL can be found in the literature on the Arcade Learning Environment (prior to DQN). For example, see "State of the Art Control of Atari Games Using Shallow Reinforcement Learning" by Liang, Machado & al. for recent work. section 5, l. 249: I would want to see this claim back up in your experiments: "far greater computational resources". The iterative SVD approach of algorithm 2. with O(n^3) per iteration is non-negligible. It is also not clear that it could benefit from the same computational speedups as GPU-based deep RL methods. section 5, l. 257: half an hour with or without the exploration phase ? section 5.1, l. 277: I find this experiment very interesting. However, if the simulator had to be modified, it suggests that the proposed approach cannot cope with the non-Markovianity by building appropriate features. Have you tried concatenating a number of successive frames (4) as in DQN ? section 5.1, l. 286: how did you choose $k$ ? section 5.1: It would be important to remind the reader that you use the planning approach of algorithm 1 in this experiment. You could have also chosen to learn the model using algorithm 2 and plan using any other approach. section 5.2, l. 321: Why did you have to define this metric rather than reporting the actual return ? section 5.2, l. 347: What was this threshold ? How did you choose it ?

Confidence in this Review

2-Confident (read it all; understood it all reasonably well)